# Inelastic Strength for Fire Resistance of Composite I-Beam Covered by Insulation Material Subjected to Basic Loading Condition

**Xuan Tung Nguyen** [ID] **and Jong Sup Park** *[ID]

Department of Civil Engineering, Sangmyung University, Cheonan 31066, Korea; tungutc2@gmail.com
* Correspondence: jonpark@smu.ac.kr; Tel.: +82-41-550-5314; Fax: 82-41-558-1201

**Abstract:** This paper presents a nonlinear numerical study on the moment resistance of composite steel-concrete beam using fire insulation subjected to various fire scenarios and basic loading conditions. The temperature-dependent material properties of fire insulation, concrete and steel were taken into consideration. The nonlinear finite element analysis was done by utilizing a commercial finite element program, ABAQUS. The obtained moment capacity of the composite I-beam from the current fire code was also performed and compared. The results showed that the fire scenarios and the fire insulation thickness have a great influence on the temperature distribution and strength degradation of the composite beam. The capacity of the beam in hydrocarbon fires, which is the most severe scenario, decreases faster than that in ISO834 standard fire and external fire. The fire resistance of the beam increases as the fire insulation thickness increases due to the temperature degradation in the steel beam. The calculated results from the current fire codes give conservative value at normal temperature and low temperature. The current fire codes can give unconservative values at high temperature when there is a great temperature discrepancy between parts of the beam. A new factor was proposed to determine the fire moment resistance of the composite beam with non-uniform temperature.

**Keywords:** nonlinear analysis; composite beam; fire; fire insulation; inelastic strength

## 1. Introduction

The steel members have been widely used in bridges and buildings because it has high strength, light weight and high aesthetics. However, the strength and stiffness of steel degrade rapidly at elevated temperatures leading to reduced loading capacity and eventually structural failure. Many experimental and numerical studies showed that the unprotected composite steel-concrete members collapse after 20–30 min of exposure to fire [1–6]. Thus, the response of steel and composite members under fire has been concerned in recent years. Besides conducting experiments, the numerical analysis method has provided effective support for the study of the behavior of the steel and composite members under normal and high temperature. Sucharda et al. [7] presented a non-linear numerical analysis of reinforced concrete beams based on the test data from Bresler and Scordelis [8], and Scordelis and Vecchio [9]. The response of reinforced concrete members under heating and cooling phases was also investigated by experiment and numerical simulation [10–12]. Test beams were simultaneously loaded and heated according to fire scenarios. After that, the structural responses during the heating and cooling phase were analyzed.

Selden et al. [13] carried out an experimental study on the thermal and structural behavior of unprotected composite beams with shear connections exposed to fire. They presented a sequentially coupled thermal-structural modeling using ABAQUS to investigate the behavior of composite beams [14]. Pak et al. [15] undertook a numerical and experimental investigation on the behavior of unprotected composite beams exposed to fire.

The plastic-damage model was used to predict the response of concrete slabs. A method to determine the stiffness degradation parameter through a nonlinear regression analysis of concrete test data was proposed. The behavior of fire exposed steel girders under both combined effects of flexural and shear loading was also investigated [16–19]. These studies indicated that shear capacity in steel beams can decrease faster than that of moment capacity because the web temperature in steel beams increases faster than flanges temperature due to the higher slenderness of the web. Kodur and Naser [20] presented a simplified approach to evaluate the shear capacity of the steel and composite beam under fire.

In order to increase the fire resistance of composite steel-concrete beams, synthetic fiber and steel fiber were used to reinforce concrete [21–24]. Besides this, fire insulation (FI) was used to cover the surface of steel beams. Kodur et al. [25] undertook finite element analysis for the effects of fire scenario and fire insulation on composite beams. The fire behavior of composite beams was investigated by applying simultaneous uniformly distributed load and fire exposure. The beam with fire insulation layers of 12.5 mm and 25 mm thickness collapsed at 60 and 120 min, respectively. Meanwhile, the failure of the beam without fire insulation occurred at 21 min. Kang et al. [26] experimentally and numerically studied the fire resistance of corrugated webbed prestressed composite beam covered by fire insulation. The insulation significantly increases the fire resistance of the beam with fire ratings over 180 min. Kodur et al. [16,27] presented strategies to enhance the fire resistance of steel bridges. They also stated that the beam using fire insulation can improve the failure time to 60–120 min. Kang et al. [28] conducted a numerical study on behavior of composite beam exposed to fire to propose a damage index for the damage judgment. Martinez and Jeffers [29] numerically investigated the behavior of restrained composite beam exposed to fire. A macro-modeling approach was proposed to predict the response of restrained composite beam under fire.

These experimental and numerical simulation studies were conducted based on the method for verification of the fire resistance of structures in the time domain or the temperature domain. It means that the structural member is subjected to simultaneous loads and fire exposure. The load remains stable but the temperature changes in the time domain method until the members fail or the test is finished. The result of this method is the failure time or the failure temperature of the structural member. Meanwhile, the load capacity of structural members at a certain time of the fire corresponding to a specific temperature is determined from heat transfer analysis and structural analysis in the strength domain method. This study aims to investigate the plastic moment capacity degradation of the composite beam with fire insulation under fire. The simply supported composite I-beam was subjected to a concentrated load at midspan and uniformly distributed load on the concrete slab. The moment capacity of the composite beam was analyzed by using a finite element program ABAQUS version 2020 (accessed on 20 January 2021) [30]. The results from the numerical simulation were compared to Eurocode 4, Part 1-2 [31] and ANSI/AISC 360-16 [32].

## 2. Moment Resistance of Composite Beams

The moment resistance of the composite beam in the positive moment can be determined by Eurocode 4, Part 1-1 [33] in the ultimate limit states. The moment resistance can be calculated by using plastic theory for Class 1 and Class 2 of steel cross-sections. Eurocode 4, Part 1-1 [33] presents that the plastic resistance moment ($M_{Pl,Rd}$) of the composite beam depends on the plastic neutral axis (PNA) position which is determined from the equilibrium equation between compressive and tensile forces in the composite section. The PNA can be located in the concrete slab, in the web, or in the top flange of the steel beam. Figure 1 shows a typical plastic stress distributions for a composite beam in the case of PNA within the concrete slab with a fully composite action. The plastic resistance moment of a composite beam at normal temperature is calculated as

$$M_{Pl,Rd} = N_{c,f}d_c + N_{pl,s}d_s \tag{1}$$

where $N_{c,f} = 0.85 f_c A_c$ is the design value of the compressive normal force in the concrete slab with fully composite action; $N_{pl,s} = f_{yd} A_s$ is the design value of the plastic moment of the steel beam section to normal force; $d_c$ and $d_s$ are the distances from the PNA to the centroids of compressive concrete slab and steel beam section, respectively; $A_c$ and $A_s$ are the effective area of the concrete slab and area of steel cross-section, respectively.

The plastic resistance moment of the composite beam under fire ($M_{fi,t,Rd}$) based on Eurocode 4, Part 1-2 [31] can be determined according to plastic theory and is given by

$$M_{fi,t,Rd} = \sum_{i=1}^{n} A_i d_i k_{y,i} f_{y,i} + 0.85 \sum_{i=1}^{n} A_j d_j k_{c,j} f_{c,j} \tag{2}$$

where $A_i$ and $A_j$ are the areas of the steel element and compressed concrete element, respectively; $d_i$ and $d_j$ are the distances from the PNA to the centroids of $A_i$ and $A_j$, respectively; $k_{y,i}$ and $k_{c,j}$ are the reduction factor in yield stress of steel and compressive strength of concrete at elevated temperature, respectively; $f_{y,i}$ is the yield stress at normal temperature for $A_i$ taken as negative on the tension side of the PNA and positive on the compression side; $f_{c,j}$ is the compressive strength at normal temperature for $A_j$.

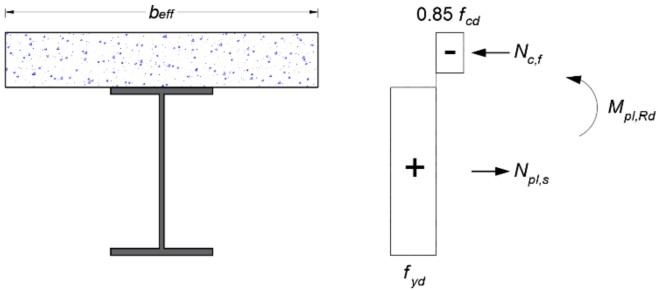

**Figure 1.** Typical plastic stress distributions for a composite beam in positive bending.

A retention factor ($r_T$) based on the bottom flange temperature was used in ANSI/AISC 360-16 [32] to calculate the flexural capacity of a composite beam under elevated temperature.

$$M_T = r_T . M_0 \tag{3}$$

where $r_T$ is the retention factor, as taken in Table 1; $M_0$ and $M_T$ are the flexural capacity of the composite beam at normal temperature and elevated temperature, respectively.

**Table 1.** Retention factor [32].

| Bottom Flange Temperature (°C) | $r_T$ |
|:---:|:---:|
| 20 | 1.00 |
| 150 | 0.98 |
| 320 | 0.95 |
| 430 | 0.89 |
| 540 | 0.71 |
| 650 | 0.49 |
| 760 | 0.26 |
| 870 | 0.12 |
| 980 | 0.05 |
| 1100 | 0.00 |

## 3. Finite Element Analysis

The finite element program ABAQUS [18] was utilized to analyze the behavior of the composite beam under fire. Figure 2 illustrates the analysis procedure which was conducted in two steps: (1) heat transfer analysis and (2) structural analysis. This process is commonly used to analyze the behavior of structures under fire [2,25,29,34,35]. The

temperature in the member obtained from the heat transfer analysis was applied to the model in structural analysis.

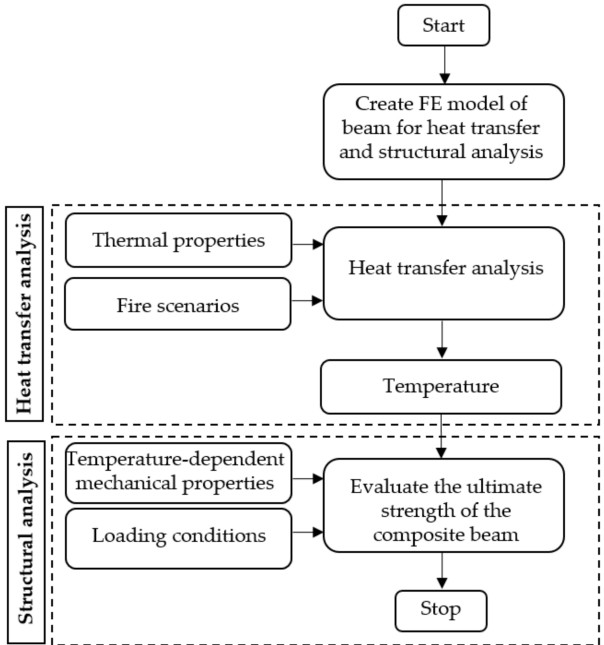

**Figure 2.** Chart of two steps in undertaking ultimate strength analysis.

*3.1. Heat Transfer Analysis*

The heat transfer analysis was conducted for composite beams with or without fire insulation (FI). The heat flux, $\dot{h}$ (W/m$^2$), from fire transfers to the beam surface through convection and radiation, can be determined by Eurocode 1, Part 1-2 [36].

$$\dot{h} = \alpha_c \left( T_f - T_m \right) + \sigma \varepsilon_f \varepsilon_m \left[ \left( T_f + 273 \right)^4 - \left( T_m + 273 \right)^4 \right] \tag{4}$$

where $\alpha_c$ is the convection coefficient and values for ISO-834 fire, external fire and hydrocarbon fire are 25 W/(m$^2$K), 25 W/(m$^2$K) and 50 W/(m$^2$K), respectively; $T_f$ and $T_m$ are the fire temperature and the surface temperature of the member (°C); $\sigma$ is the Stefan-Boltzmann constant ($\sigma = 5.67 \times 10^{-8}$ W/(m$^2$K$^4$)); $\varepsilon_f$ is the emissivity of the fire and is taken as 1.0 based on the recommendation from Eurocode 1, part 1-2 [36]; $\varepsilon_m$ is the surface emissivity of the member. Due to the influence of the depth of the beam, the surface emissivity of parts in the composite beam increases with vertical distance from top to bottom of the beam. The emissivity factors of the slab, top flange, web, and bottom flange are 0.3, 0.3, 0.5 and 0.7, respectively, based on previous studies [2,25]. In heat transfer analysis, the four-node heat transfer quadrilateral shell element (DS4) was used to model the steel beam. The slab concrete and fire insulation were modeled with eight-node linear heat transfer brick element (DC3D8). The reinforcing steel was simulated using two-node link (DC1D2). The temperature-dependent thermal properties that included specific heat, thermal conductivity and density were introduced into the heat transfer model.

*3.2. Structural Analysis*

The temperatures obtained from thermal analysis were used to impose to the model in the structural analysis. The mesh size of the model was divided the same as in the thermal analysis. The fire insulation layer was not taken into account in this step. The slab concrete, steel beam and reinforcing steel were modeled utilizing an eight-node linear brick element with reduced integration (C3D8R), the four-node reduced integration shell element (S4R), and a two-node linear three-dimensional truss (T3D2), respectively. The interaction

between reinforcing steel and the concrete slab was implemented by using embedded region. To consider the fully composite action between the steel beam and the concrete slab, multi-point constraints (MPC) are used to link nodes of shell element and to nodes of the solid element. The moment capacity of the composite beam was predicted by using the Riks method, which uses the load magnitude as an additional unknown and solves simultaneously for loads and displacements. This method is available in ABAQUS [30]. Figure 3 shows the typical three-dimensional meshing and boundary condition of the simply supported composite beam. Both tips of the bottom flange are restricted against displacements in X and Y directions. One tip is free to move in the Z-direction for the roller support and another one is restricted against displacements in the Z-direction for the hinge support. A convergence study was done to find the optimum mesh size for steel beam and concrete slab models. Based on the results, elements size of 50 × 50 mm and 50 × 50 × 50 mm were adopted for steel beam and concrete slab models, respectively.

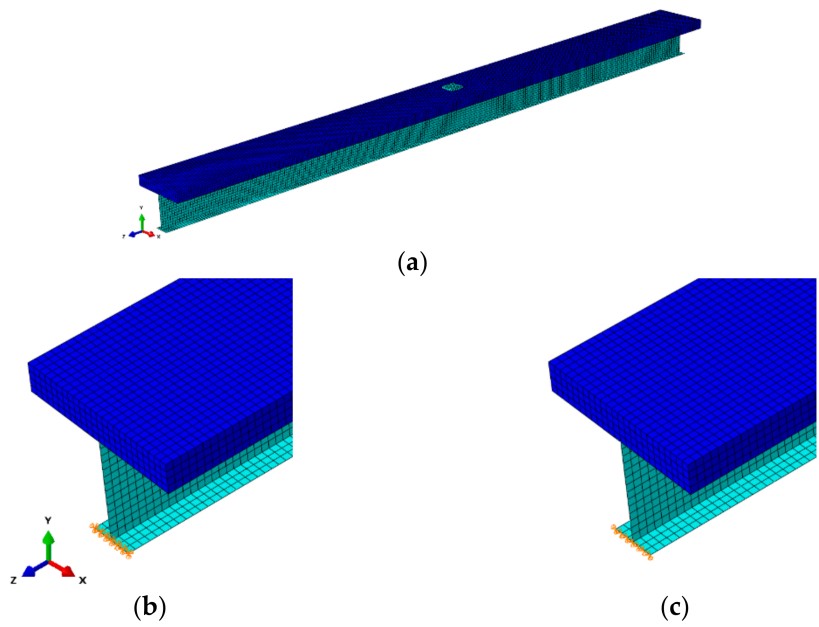

(a)

(b)                                                    (c)

**Figure 3.** Finite element models and boundary condition: (**a**) Three-dimensional meshing; (**b**) Hinge support; (**c**) Roller support.

### 3.3. Material Properties

The temperature-dependent material properties of concrete, steel and fire insulation must be considered to analyze the behavior of the composite beam under fire conditions. The properties at high temperature of concrete and steel were obtained according to Eurocode 2, Part 1-2 [37] and Eurocode 3, Part 1-2 [38], respectively. The thermal properties of steel were introduced into the heat transfer model including specific heat and thermal conductivity as shown in Figure 4a–d showed thermal elongation coefficient, yield strength, proportional limit, elastic modulus and the stress-strain relationship of the steel.

Figure 5a,b shows the specific heat and thermal conductivity of the concrete, respectively. The temperature-dependent mechanical properties of concrete include thermal elongation coefficient, compressive stress-strain relationship as shown in Figure 5c,d. The thermal elongation coefficient of concrete was shown in Figure 4b. The fire insulation used in this study is CAFCO 300, which is widely employed to enhance the fire resistance of structure [25,29,39–41]. Temperature-dependent thermal properties of CAFCO 300 were taken from the experiment conducted by Imran et al. [41]. Figure 6 shows the density, thermal conductivity, and specific heat of CAFCO 300 as a function of temperature.

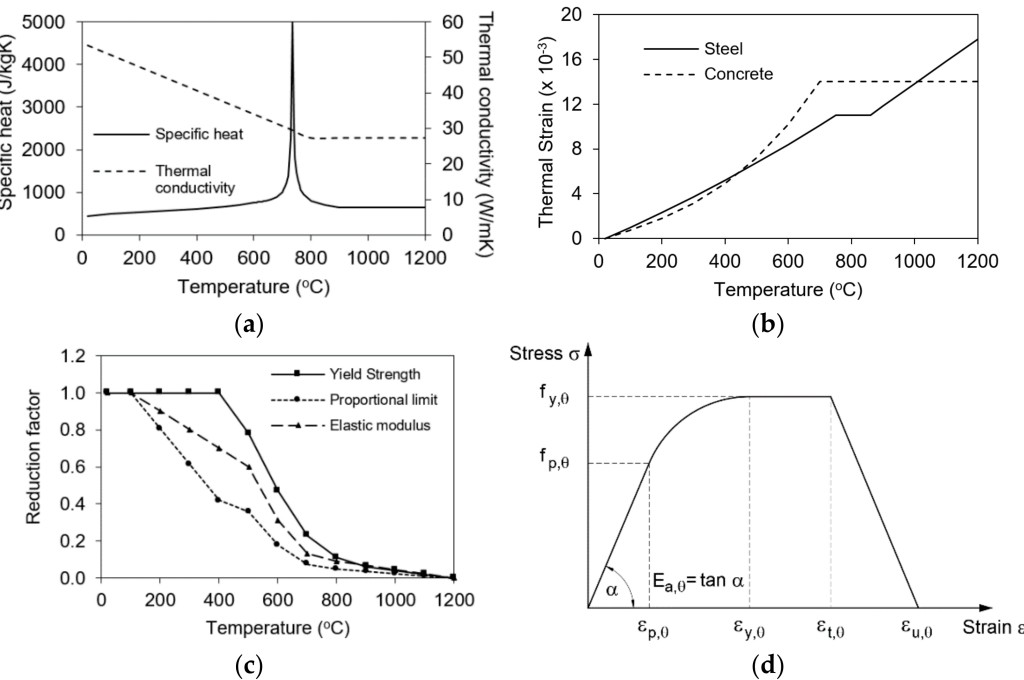

**Figure 4.** Material properties of steel at high temperature: (**a**) Specific heat and thermal conductivity; (**b**) Thermal elongation; (**c**) Strength and stiffness reduction factor; (**d**) Stress-strain curve at elevated temperature.

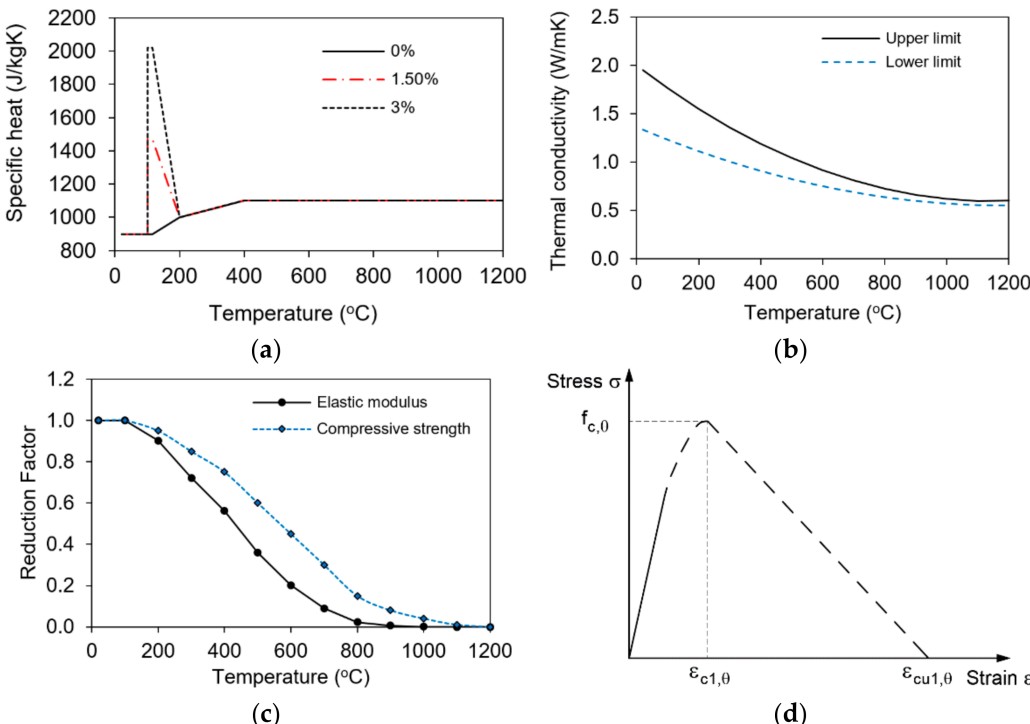

**Figure 5.** Material properties of concrete at high temperature: (**a**) Specific heat; (**b**) Thermal conductivity; (**c**) Strength and stiffness reduction factor; (**d**) Stress-strain curve at elevated temperature.

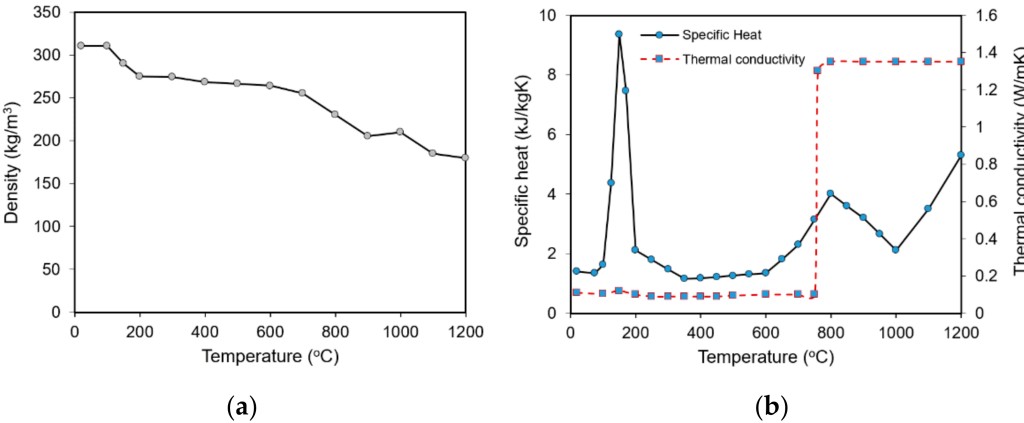

**Figure 6.** Material properties of insulation at high temperature: (**a**) Density; (**b**) Specific heat and thermal conductivity.

## 4. Validation of Model

The verification of the model was performed by comparing finite element analysis (FEA) results with test data of the composite steel-concrete beam conducted by Wainman and Kirby [42]. The beam was subjected to four-point loads and ISO-834 fire [43] exposure until failure. Figure 7a illustrates the boundary and loading conditions of the tested beam and Figure 7b shows the detail sectional dimensions. This composite beam includes a steel beam with a yield stress of 255 MPa and a concrete slab with a compressive strength of 30 MPa.

Figure 8 presents a comparison between FEA results and test data. Figure 8a presents that the temperatures in the steel beam obtained from FEA results are in good agreement with that from test data. Figure 8b shows the comparison of midspan deflection between FEA and test data. The deflection at midspan increased with time exposed to fire and the beam failed at 22.5 min. The FEA results provided good agreement with the test data with a predicted failure time of 22 min. The results indicated that the finite element model gave reliable results to investigate the fire resistance of composite beam under fire.

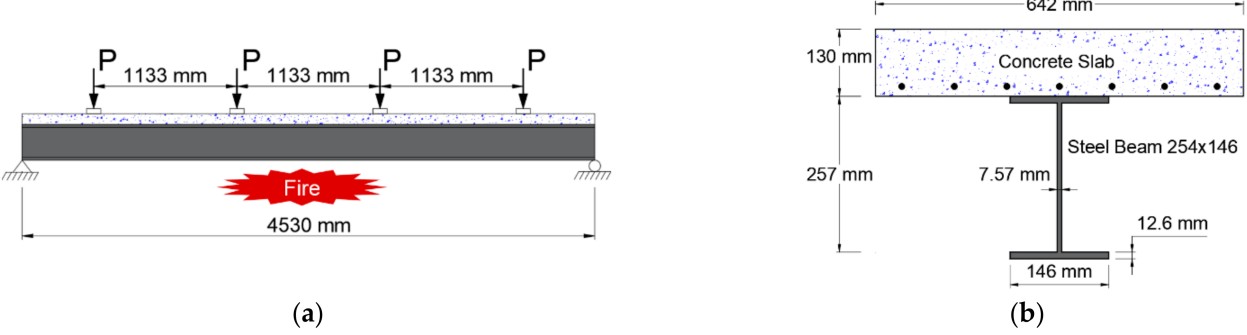

**Figure 7.** Tested beam layout used in validation model: (**a**) Boundary and loading conditions; (**b**) Cross-section.

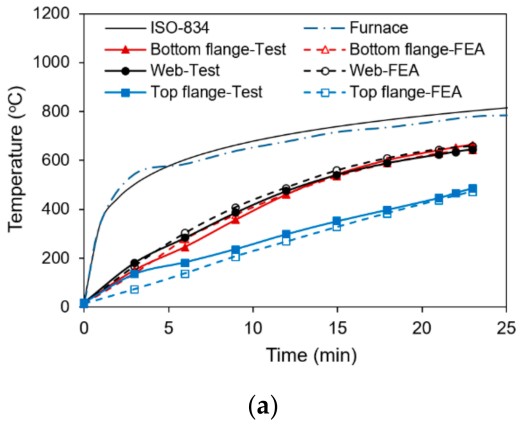 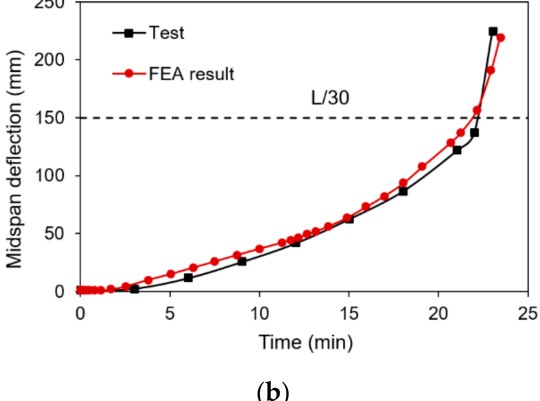

|(a)|(b)|

**Figure 8.** Comparison of FEA results and test data: (**a**) Temperature in steel beam; (**b**) Midspan deflection during fire.

## 5. Model Properties

A steel I-beam with M700 × 300 cross-section from Hyundai Steel [44] supporting a concrete slab with 200 mm thickness was selected to investigate the fire resistance of the simply supported composite beam under fire. Tables 2 and 3 present the detailed properties of the steel beam and concrete slab under normal temperature and elevated temperature, respectively. Load cases include a concentrated load at midspan (LC1) and uniformly distributed load on the concrete slab (LC2). Figure 9a shows the loading conditions and the clear span length of 14 m. Figure 9b illustrates the cross-section of the composite beam. Longitudinal reinforcing steels with a diameter of 14 mm were used in the concrete slab.

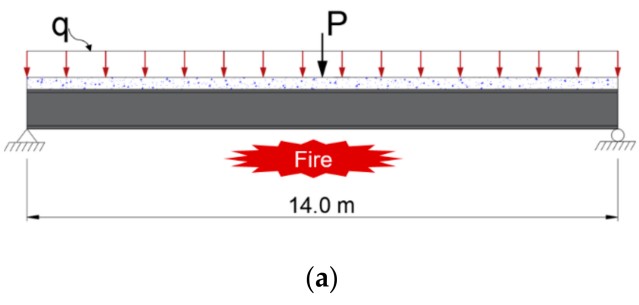 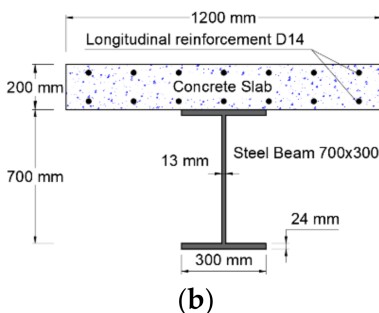

|(a)|(b)|

**Figure 9.** Composite beam used in analysis: (**a**) Boundary and loading conditions; (**b**) Cross-section of composite beam.

**Table 2.** Properties of composite beam.

| Types | Properties | Values |
|---|---|---|
| Steel beam | Beam height, $h$ (mm) | 700 |
| | Flange width, $b_f$ (mm) | 300 |
| | Flange thickness, $t_f$ (mm) | 24 |
| | Web thickness, $t_w$ (mm) | 13 |
| | Elastic modulus, $E$ (GPa) | 210 |
| | Yield stress, $f_y$ (MPa) | 275 |
| | Poisson's ratio, $\mu_s$ | 0.3 |
| Concrete slab | Slab thickness, $t_s$ (mm) | 200 |
| | Slab width, $b_c$ (mm) | 1200 |
| | Compressive strength, $f_c$ (MPa) | 30 |
| | Poisson's ratio, $\mu_c$ | 0.2 |

Fire insulation with the thickness of 10 mm, 20 mm and 30 mm was used to cover the steel beam to investigate the influence of fire insulation on the behavior of the composite beam as shown Figure 10. The influence of fire scenarios on the behavior of the composite

beam was also taken into account. Figure 11 shows the temperature-time curves for the various fire scenarios; hydrocarbon (HC) fire, ISO-834 and external fire.

**Table 3.** Properties of steel and concrete at elevated temperature.

| Temperature (°C) | Steel Beam | | Concrete Slab | |
|---|---|---|---|---|
| | Yield Stress, $f_y$ (MPa) | Elastic Modulus, E (MPa) | Compressive Strength, $f_c$ (MPa) | Elastic Modulus, E (MPa) |
| 20 | 275.00 | 210,000 | 30.00 | 30,588.56 |
| 100 | 275.00 | 210,000 | 30.00 | 30,588.56 |
| 200 | 275.00 | 189,000 | 28.50 | 27,606.18 |
| 300 | 275.00 | 168,000 | 25.50 | 22,100.24 |
| 400 | 275.00 | 147,000 | 22.50 | 17,206.07 |
| 500 | 214.50 | 126,000 | 18.00 | 11,011.88 |
| 600 | 129.25 | 65,100 | 13.50 | 6194.18 |
| 700 | 63.25 | 27,300 | 9.00 | 2752.97 |
| 800 | 30.25 | 18,900 | 4.50 | 688.24 |
| 900 | 16.50 | 14,175 | 2.40 | 195.77 |
| 1000 | 11.00 | 9450 | 1.20 | 48.94 |

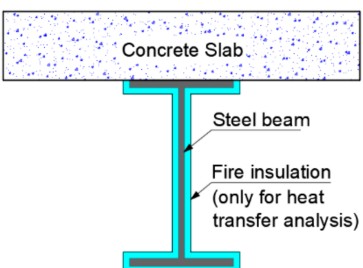

**Figure 10.** Protected steel beam using insulation.

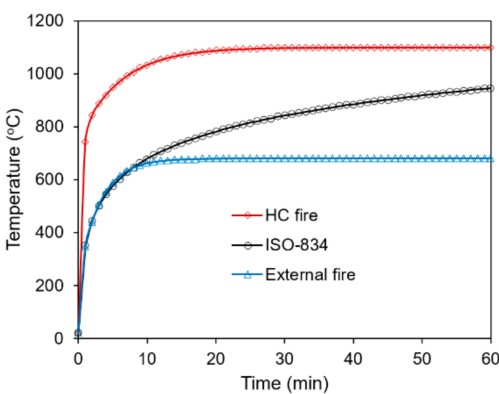

**Figure 11.** Various fire scenarios.

## 6. Finite Element Analysis Results

### 6.1. Temperature Distribution in the Beam

Table 4 illustrated the temperature distributions in the composite beams subjected to fire. The data show that the temperature distributions in the steel beam and the concrete slab are non-uniform on the depth of beam cross-section. Due to the effect of the fire insulation, this non-uniform distribution of temperature in the steel beam becomes more evident as the fire insulation thickness increases. The temperature in the steel beam using FI decreases as the FI thickness increase. The temperature in the concrete slab is much lower than the temperature in the steel beam because the concrete slab has a large thickness as well as its low thermal conductivity and higher specific heat compared to the steel beam. Figure 12a–c indicate the temperature-time graphs of the steel beam subjected to ISO-834 fire, external fire and hydrocarbon fire, respectively. The temperature of the top flange in

all cases is much smaller than the temperature in the web and the bottom flange because the top flange was protected by the concrete slab with lower thermal conductivity and higher thermal capacity. The web temperature is slightly higher than the bottom flange temperature because the web thickness is thinner and the web surface area exposed to fire is larger than the flange. In the case of without FI, the temperatures in the bottom flange and web increase gradually with the increasing pace of the fire temperature and go asymptotically to the peak temperature of the fire. The maximum temperatures in the steel beam are 938 °C in ISO-834 fire, 679 °C in external fire, and 1100 °C in HC fire at 60 min.

The insulation has a great effect on temperature in steel beams. In ISO-834 fire and external fire, which are less severe than HC fire, the effect of insulation is very good. In the case of ISO-834, the temperature in the steel beam increases slowly after 40 min and the temperature in the steel beam is less than 400 °C at which the yield strength of steel remains unchanged compared to the normal temperature, as shown in Figure 4c. At 60 min, the maximum temperatures in the steel beam are 720 °C with 10-mm-thickness FI, 451 °C with 20-mm-thickness FI and 270 °C with 30-mm-thickness FI. The temperature in the steel beam also increases slowly and is less than 400 °C after 60 min in the case of external fire. In the most severe fire scenario of the HC fire, the temperature in the steel beam is higher than that in the cases of ISO-834 and external fire. At 60 min, the maximum temperatures in the steel beam are 1091 °C with 10-mm-thickness FI, 1027 °C with 20-mm-thickness FI, and 870 °C with 30-mm-thickness FI, respectively.

**Table 4.** Temperature distribution in ISO-834 fire at 30 min.

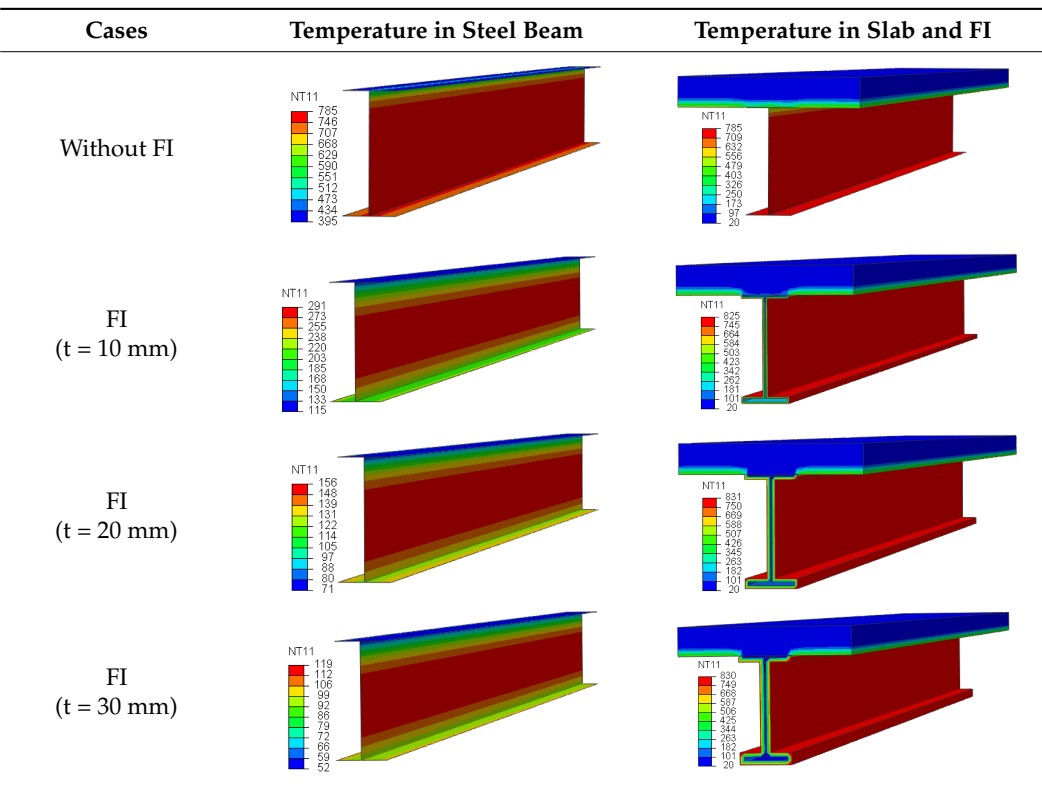

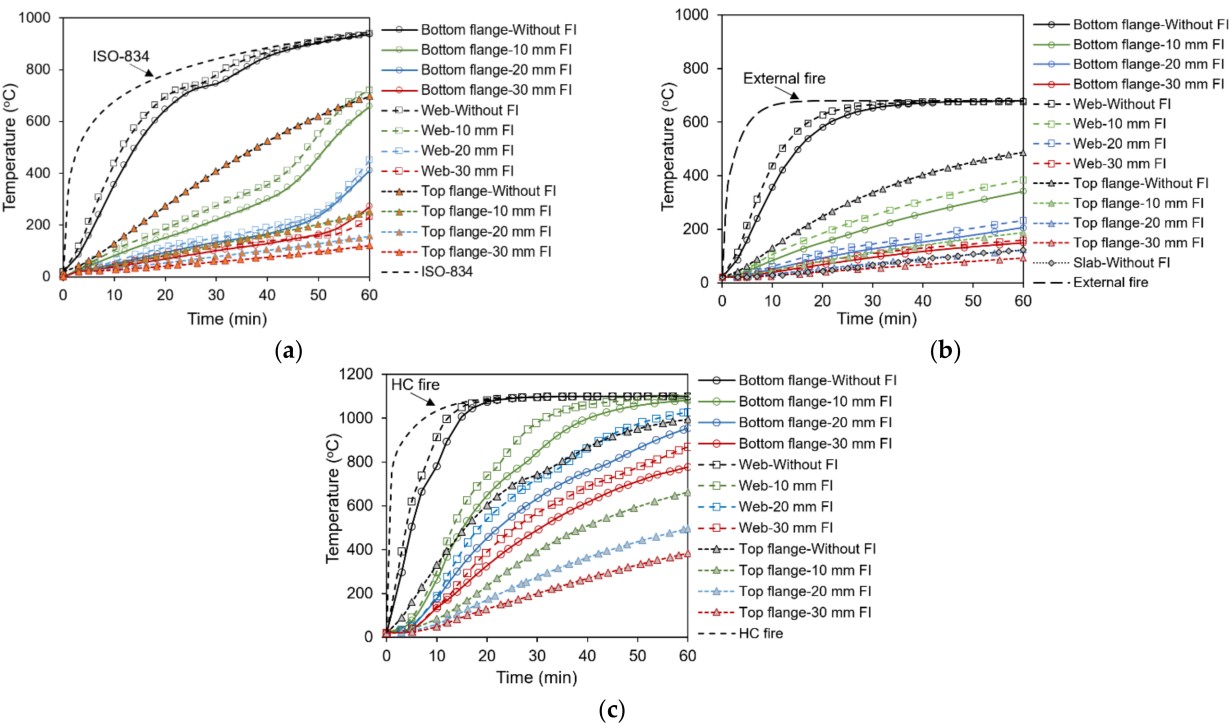

**Figure 12.** Temperature distribution in steel beam: (**a**) ISO-834 fire; (**b**) External fire; (**c**) Hydrocarbon fire.

### 6.2. Influence of Fire Scenarios and Load Cases on the Moment Resistance

The influence of fire scenarios on the bending capacity of the composite beam is investigated with three different fire scenarios, namely, ISO-834, external fire and HC fire. Figure 13 presents the reduction ratio in the bending capacity of the composite beam without FI. The FEA results were plotted as the ratio of the bending capacity of the composite beam at fire exposure time ($M_T$) to the bending capacity at normal temperature ($M_0$). The fastest decreasing in the moment capacity of the composite beam was observed in HC fire. The moment resistance degrades rapidly to 22% after 10 min of HC fire exposure and then decreases gradually to 5% at 60 min for LC1. Meanwhile, the moment resistance of the composite beam reduces to 41% and 56% at 20 min in the case of ISO-834 and external fire, respectively. After that, the moment resistance of the composite beam decreases steadily to 11% at 60 min in the case of ISO-834, whereas the moment capacity degrades gradually to 37% at 60 min in the case of external fire. Figure 13 also shows that the bending capacity of the beam under fire with a uniformly distributed load (LC2) decreases more than that with the concentrated load (LC1). The bending capacity of the beam with LC2 degrades to 3%, 9% and 36% at 60 min in the cases of HC fire, ISO-8324 and external fire, respectively.

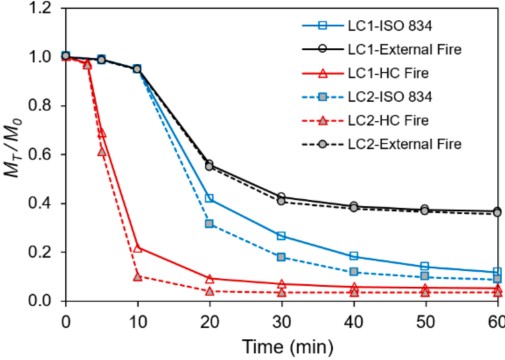

**Figure 13.** Reduction ratio of the composite beam without fire insulation.

### 6.3. Influence of Fire Insulation Thickness on the Moment Resistance

To investigate the influence of FI thickness on the fire resistance of composite beam, the FI with thicknesses of 10, 20 and 30 mm was used to protect steel beams. Figure 14 shows the reduction ratio in moment capacity of composite beam with LC1. The FEA results of Figure 14 indicate that the moment resistance of the composite beam using FI is greater than that in the case of without FI and the capacity of the beam using FI increases as the FI thickness increases because the temperature in the steel beam using FI is much smaller than that in the case of without FI and the temperature in the beam decreases as the FI thickness increases.

Figure 14a shows the reduction ratio in the moment capacity of the composite beam with and without FI in ISO-834 fire. In the case of beams using FI, the thickness of FI has a negligible effect on the fire resistance of the protected composite beam at 40 min. The moment capacity decreases gradually to 41% with 10-mm-thickness FI, 92% with 20-mm-thickness FI, and 97% with 30-mm-thickness FI at 60 min.

In the case of external fire, the moment resistance of the composite beam using FI degrades slightly at 60 min, as shown in Figure 14b. The moment capacity of the composite beam reduces to 97% with 10-mm-thickness FI, 98% with 20-mm-thickness FI, and 99% with 30-mm-thickness FI.

In the case of HC fire, which is the most severe fire, it was also seen that the moment capacity of the beam using FI is greater than that without FI as shown in Figure 14c. In the case of beams using FI, the thickness of FI has a significant effect on the fire resistance of the composite beam. The moment capacity decreases quickly to 8% with 10-mm-thickness FI, 12% with 20-mm-thickness FI, and 23% with 30-mm-thickness FI at 60 min.

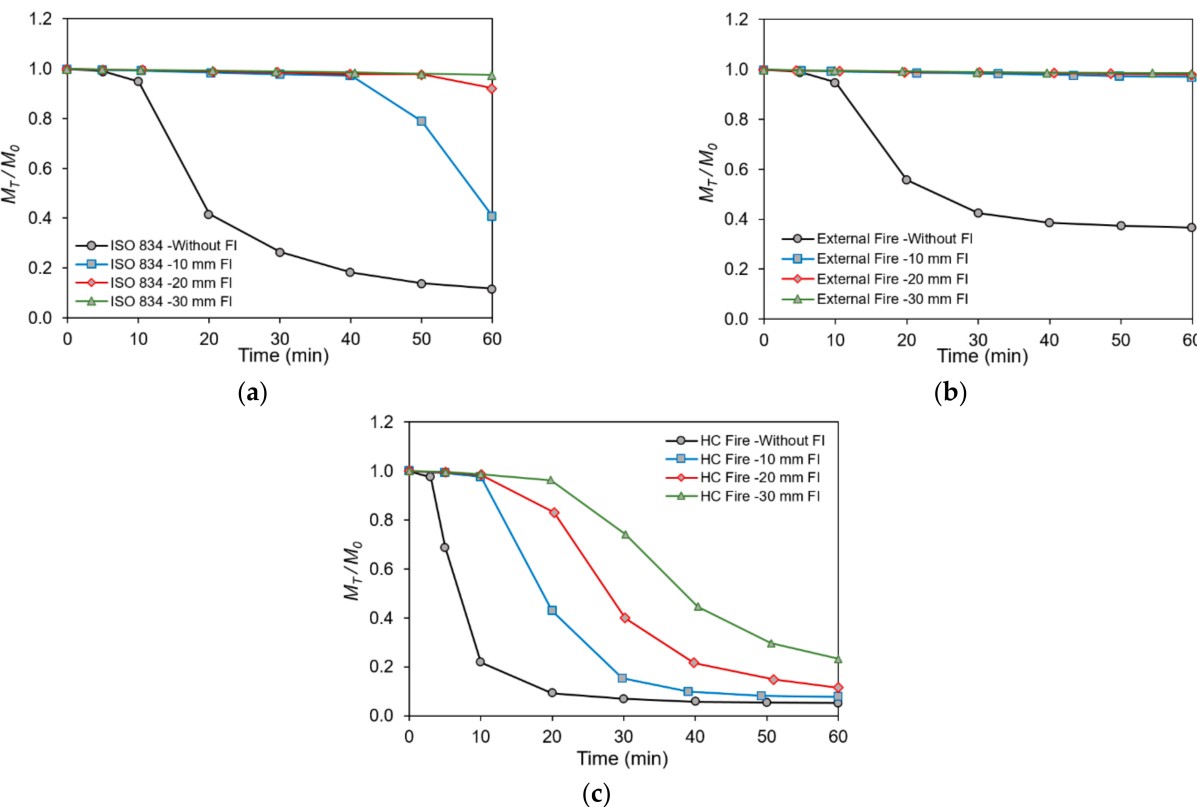

**Figure 14.** Reduction ratio of the composite beam with LC1: (**a**) ISO-834 fire; (**b**) External fire; (**c**) Hydrocarbon fire.

### 6.4. Comparison with Design Codes

A comparison between the FEA results and Eurocode 4, Part 1-2 [31] and ANSI/AISC [32] was done for the composite beam with and without FI subjected to fire. The results obtained

from FEA, Eurocode 4 and ANSI/AISC are presented in Table A1 (see Appendix A). In Figure 15, the ratios of the FEA results to Eurocode 4 and ANSI/AISC are plotted against the bottom flange temperature in the steel beam. The results indicate that the FEA results give good agreement with both design codes when the bottom flange temperature in the steel beam is less than 400 °C. It can also be observed that the moment resistance of the composite beam using Eurocode 4 and ANSI/AISC can give unconservative values when the maximum temperature in the steel beam is greater than 400 °C. In the case of using Eurocode 4, the maximum differences for an unconservative estimate are 41.2% and 309.1% for LC1 and LC2, respectively. In the case of using ANSI/AISC, the maximum differences for an unconservative estimate are 26.4% and 167% for LC1 and LC2, respectively. This is because there is a large temperature difference between the parts of the beam, as shown in Figure 12. Therefore, the strength and stiffness of the top flange are much greater than those of the bottom flange and web. The top flange can be not stressed to its yield strength when the beam fails. ANSI/AISC recommends that the temperature should be taken as constant between the bottom flange and the web and the temperature difference between the web and the top flange of the beam is not more than 25%. Besides this, the moment resistance of the composite beam using Eurocode 4 is based on plastic theory and the steel members are stressed to their yield strength in compression or tension. Thus, as the temperature in the steel beam increases and the larger the temperature difference between the parts are, the more unconservative results are.

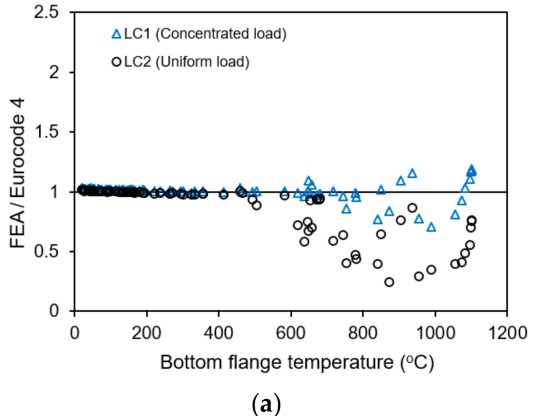
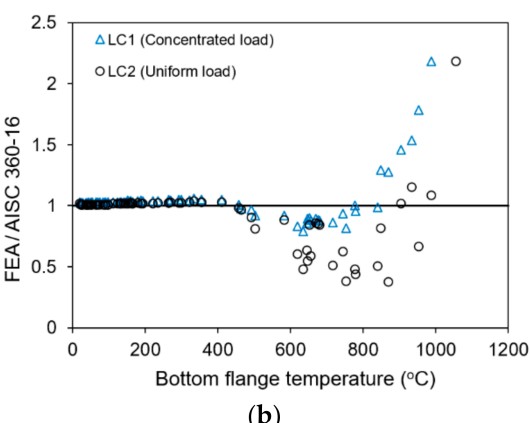

(**a**)  (**b**)

**Figure 15.** Comparison between FEA results and design codes: (**a**) Eurocode 4, Part 1-2 [31]; (**b**) AISC [32].

### 6.5. Proposal for Fire Resistance Moment

The new equation for the fire resistance moment of composite beams with non-uniform temperature is proposed based on Equation (3) and FEA results

$$M_{fi} = \frac{r_T}{f}.M_0 \qquad (5)$$

where $r_T$ is taken in Table 5; $M_0$ and $M_{fi}$ are the nominal flexural capacity at normal temperature and under fire, respectively; $f$ is the new factor for the fire resistance moment of composite beams with non-uniform temperature and is determined according to the bottom flange temperature ($T$), as given in Table 5.

**Table 5.** Factors for composite flexural beams under non-uniform temperature.

| Load Cases | Bottom Flange Temperature, $T$ ($^{\circ}$C) | $f$ | $r_T$ |
|---|---|---|---|
| LC1 | $T \leq 430$ | 1 | AISC * |
| | $430 < T \leq 650$ | $0.0016T + 0.2982$ | AISC * |
| | $650 < T \leq 800$ | $-0.0009T + 1.9771$ | AISC * |
| | $800 < T \leq 1000$ | $-0.0028T + 3.4557$ | AISC * |
| | $T = 1100$ | 1 | 0.05 |
| LC2 | $T \leq 430$ | 1 | AISC * |
| | $430 < T \leq 870$ | $0.0051T - 1.2825$ | AISC * |
| | $870 < T \leq 1000$ | $-0.0124T + 13.846$ | AISC * |
| | $T = 1100$ | 1 | 0.025 |

AISC *: $r_T$ is taken according to the AISC, as shown in Table 1.

Figure 16 shows the comparison of reduction ratio in the fire resistance moment of the composite beam between FEA results, AISC 360-16 and the proposal equation. It can be seen that the proposal for fire resistance moment provides conservative values for most cases and good agreement with FEA results. It can be seen that the proposal for fire resistance moment provides conservative values for most cases and good agreement with FEA results. Figure 17 shows the ratio of FEA results to predicted values from the proposal for both load cases. The data above the line of 1 gives conservative estimates or is safe with respect to FEA. The figure shows that the proposal gives conservative values for most cases of both load cases. The maximum differences for a conservative estimate are 47% and 62.5% for LC1 and LC2, respectively. It also shows that 2.4% of the models provide unconservative value with a maximum difference of 6.2% for LC2. Comparing Figures 15 and 17 shows that the capacity of the composite beam predicted by the using new proposal gives more reasonable accuracy and is safer.

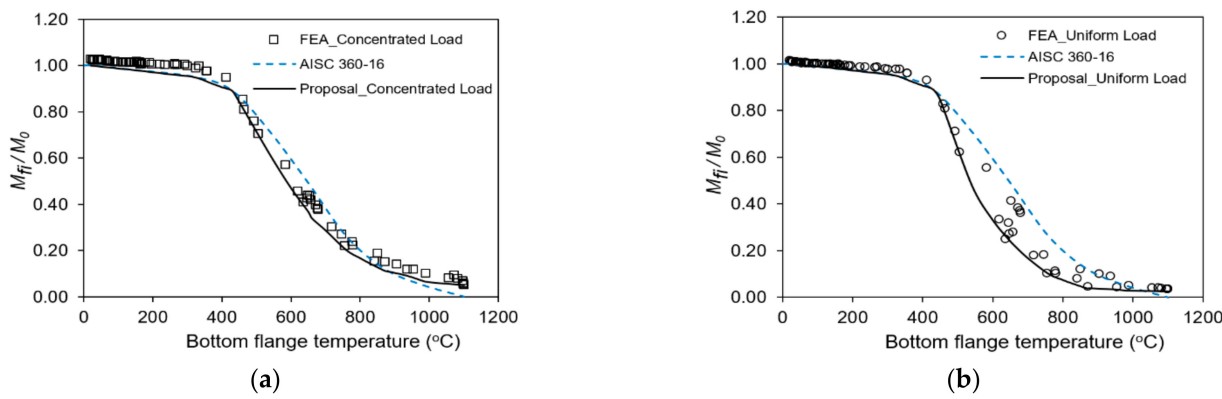

(**a**)　　　　　　　　　　　　　　　　　(**b**)

**Figure 16.** Comparison between FEA results and proposal equation: (**a**) LC1; (**b**) LC2.

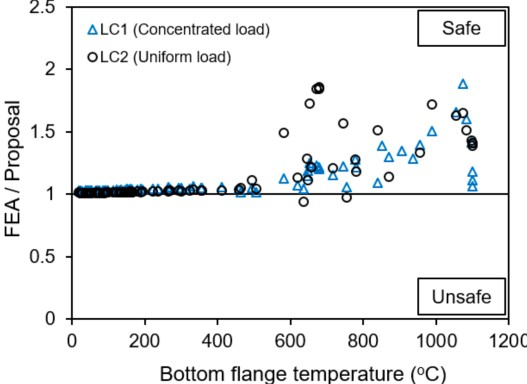

**Figure 17.** Accuracy of reduction factor obtained from FEA compared to new proposal.

## 7. Conclusions

This study presents the moment resistance of composite steel-concrete beam using fire insulation subjected to various fire scenarios and basic loading condition. The nonlinear finite element analysis was done by utilizing ABAQUS. The following conclusions can be drawn based on the results of finite element analysis:

(1) Temperature within the beam subjected to fire is non-uniformly distributed and the temperature in the steel beam using FI decreases as the FI thickness increase.

(2) The fastest decrease in the moment resistance of the composite beam was observed in HC fire. The moment resistance of the composite beam under fire with a uniformly distributed load decreases more than that with the concentrated load. The moment capacity degrades to 5% and 3% at 60 min for LC1 and LC2 in the case of without FI, respectively.

(3) The moment resistance of the composite beam using FI is greater than that in the case of without FI. In addition, the capacity of the beam using FI increases as the FI thickness increases due to the temperature degradation in the steel beam. The moment capacity of the beam with 10- and 30-mm-thickness FI decreases to 12% and 23% for LC1 in the case of HC fire, respectively. Besides this, in the external fire, which is the least severe scenario, the influence of FI thickness on fire resistance of protected composite beam is not significant in the external fire when the FI thickness increases from 10 mm to 30 mm. The moment capacity of the composite beam is almost unchanged after 60 min of exposure to external fire.

(4) The calculated moment capacities of the composite beam from Eurocode 4 and ANSI/AISC were also generated and compared to the FEA results. A good agreement is observed between FEA results and fire design codes when the bottom flange temperature in the steel beam is less than 400 °C. Nonetheless, the moment capacity of composite steel-concrete beams using both fire codes can give unconservative results when the bottom flange temperature in the steel beam is greater than 400 °C.

(5) A new factor was proposed for predicting the moment capacity of composite steel-concrete beams subjected to a concentrated load at midspan and uniformly distributed load under non-uniform heating condition. The suggested equation provides conservative values for most cases and good agreement with FEA results.

**Author Contributions:** Conceptualization, J.S.P.; methodology, X.T.N. and J.S.P.; software, X.T.N.; validation, X.T.N.; formal analysis, J.S.P. and X.T.N.; investigation, X.T.N.; resources, J.S.P.; data curation, J.S.P.; writing—original draft preparation, X.T.N.; writing—review and editing, J.S.P.; visualization, J.S.P.; supervision, J.S.P.; project administration, J.S.P.; funding acquisition, J.S.P. All authors have read and agreed to the published version of the manuscript.

**Funding:** This research was supported by the National Research Foundation (NRF) of Korea grant funded by the Korea government (Ministry of Science and ICT) (No. 2019R1F1A1060708).

**Institutional Review Board Statement:** Not applicable.

**Informed Consent Statement:** Not applicable.

**Data Availability Statement:** Data are contained within the article.

**Conflicts of Interest:** The authors declare no conflict of interest.

## Appendix A

This appendix presents Table A1 in which the results obtained from the FEA analysis and design codes are summarized.

**Table A1.** Moment capacity obtained from Eurocode 4, ANSI/AISC, FEA and proposal; unit: kN·m.

| Cases | Time, Min | Bottom Flange Temperature, °C | Eurocode 4 LC1, LC2 | AISC LC1, LC2 | FEA LC1 | FEA LC2 | Proposal LC1 | Proposal LC2 |
|---|---|---|---|---|---|---|---|---|
| ISO834-Without FI | 0 | 20 | 2819.3 | 2819.3 | 2900.1 | 2859.31 | 2819.3 | 2819.3 |
| | 5 | 158 | 2819.3 | 2758.9 | 2869.0 | 2818.03 | 2758.9 | 2758.9 |
| | 10 | 355 | 2751.0 | 2623.9 | 2752.5 | 2708.72 | 2623.9 | 2623.9 |
| | 20 | 646 | 1202.3 | 1404.4 | 1206.2 | 897.87 | 1054.6 | 698.1 |
| | 30 | 745 | 801.3 | 820.0 | 769.3 | 510.29 | 627.7 | 325.6 |
| | 40 | 850 | 521.7 | 408.9 | 529.1 | 335.60 | 380.5 | 133.9 |
| | 50 | 905 | 368.8 | 276.0 | 402.3 | 280.23 | 299.2 | 105.0 |
| | 60 | 936 | 291.9 | 219.3 | 337.1 | 252.71 | 263.0 | 98.1 |
| External fire-Without FI | 5 | 160 | 2819.3 | 2757.7 | 2869.2 | 2816.69 | 2757.7 | 2757.7 |
| | 10 | 356 | 2750.0 | 2622.5 | 2751.8 | 2707.87 | 2622.5 | 2622.5 |
| | 20 | 582 | 1610.9 | 1763.1 | 1617.5 | 1560.86 | 1433.5 | 1044.9 |
| | 30 | 652 | 1251.2 | 1371.3 | 1233.9 | 1161.45 | 986.1 | 671.8 |
| | 40 | 672 | 1158.2 | 1254.1 | 1121.3 | 1079.58 | 913.6 | 585.3 |
| | 50 | 677 | 1112.0 | 1219.6 | 1082.7 | 1042.74 | 891.9 | 561.4 |
| | 60 | 679 | 1084.6 | 1209.2 | 1065.0 | 1020.45 | 885.3 | 554.3 |
| HC fire-Without FI | 5 | 505 | 1979.8 | 2163.4 | 1991.6 | 1751.24 | 1955.8 | 1673.5 |
| | 10 | 780 | 658.7 | 662.2 | 631.6 | 290.19 | 519.2 | 245.8 |
| | 20 | 1073 | 286.3 | 31.6 | 266.0 | 116.45 | 141.0 | 70.5 |
| | 30 | 1096 | 180.9 | 4.6 | 199.7 | 100.93 | 141.0 | 70.5 |
| | 40 | 1099 | 141.8 | 0.8 | 166.2 | 99.29 | 141.0 | 70.5 |
| | 50 | 1100 | 131.5 | 0.2 | 156.4 | 99.10 | 141.0 | 70.5 |
| | 60 | 1100 | 128.1 | 0.0 | 150.4 | 98.05 | 141.0 | 70.5 |
| ISO834-10 mm FI | 5 | 42 | 2819.3 | 2809.6 | 2885.1 | 2836.86 | 2809.6 | 2809.6 |
| | 10 | 85 | 2819.3 | 2790.9 | 2876.7 | 2823.33 | 2790.9 | 2790.9 |
| | 20 | 153 | 2819.3 | 2761.3 | 2853.8 | 2805.58 | 2761.3 | 2761.3 |
| | 30 | 221 | 2819.3 | 2727.4 | 2837.1 | 2780.76 | 2727.4 | 2727.4 |
| | 40 | 300 | 2819.3 | 2688.3 | 2820.0 | 2758.08 | 2688.3 | 2688.3 |
| | 50 | 464 | 2300.6 | 2350.6 | 2290.6 | 2275.33 | 2257.5 | 2164.8 |
| | 60 | 658 | 1118.4 | 1336.1 | 1184.2 | 784.20 | 964.6 | 644.9 |
| External fire-10 mm FI | 5 | 45 | 2819.3 | 2808.5 | 2900.1 | 2835.94 | 2808.5 | 2808.5 |
| | 9 | 66 | 2819.3 | 2799.3 | 2884.5 | 2824.55 | 2799.3 | 2799.3 |
| | 21 | 151 | 2819.3 | 2762.2 | 2877.7 | 2807.03 | 2762.2 | 2762.2 |
| | 29 | 189 | 2819.3 | 2743.5 | 2861.9 | 2794.46 | 2743.5 | 2743.5 |
| | 43 | 265 | 2819.3 | 2705.9 | 2852.9 | 2778.78 | 2705.9 | 2705.9 |
| | 50 | 293 | 2819.3 | 2691.8 | 2835.4 | 2764.42 | 2691.8 | 2691.8 |
| | 60 | 334 | 2817.4 | 2656.9 | 2819.6 | 2754.25 | 2656.9 | 2656.9 |
| HC fire-10 mm FI | 5 | 75 | 2819.3 | 2795.5 | 2876.8 | 2826.11 | 2795.5 | 2795.5 |
| | 10 | 265 | 2819.3 | 2705.8 | 2828.6 | 2776.21 | 2705.8 | 2705.8 |
| | 20 | 648 | 1138.4 | 1393.2 | 1241.3 | 765.41 | 1043.7 | 689.1 |
| | 30 | 840 | 570.7 | 445.2 | 440.4 | 224.35 | 403.6 | 148.3 |
| | 39 | 988 | 404.9 | 131.2 | 286.9 | 142.10 | 190.6 | 82.5 |
| | 49 | 1055 | 288.4 | 52.6 | 233.9 | 115.04 | 141.0 | 70.5 |
| | 60 | 1082 | 219.4 | 21.1 | 226.0 | 106.86 | 141.0 | 70.5 |
| ISO834-20 mm FI | 5 | 29 | 2819.3 | 2815.3 | 2889.4 | 2846.56 | 2815.3 | 2815.3 |
| | 11 | 53 | 2819.3 | 2805.0 | 2883.5 | 2837.19 | 2805.0 | 2805.0 |
| | 21 | 98 | 2819.3 | 2785.6 | 2869.5 | 2822.49 | 2785.6 | 2785.6 |
| | 30 | 133 | 2819.3 | 2770.2 | 2859.4 | 2811.07 | 2770.2 | 2770.2 |
| | 40 | 166 | 2819.3 | 2754.8 | 2838.0 | 2801.11 | 2754.8 | 2754.8 |
| | 50 | 236 | 2817.7 | 2720.1 | 2831.8 | 2782.86 | 2720.1 | 2720.1 |
| | 60 | 412 | 2686.4 | 2537.1 | 2676.1 | 2619.82 | 2537.1 | 2537.1 |

**Table A1.** *Cont.*

| Cases | Time, Min | Bottom Flange Temperature, °C | Eurocode 4 | AISC | FEA | | Proposal | |
|---|---|---|---|---|---|---|---|---|
| | | | LC1, LC2 | LC1, LC2 | LC1 | LC2 | LC1 | LC2 |
| External fire-20 mm FI | 5 | 27 | 2819.3 | 2816.1 | 2887.5 | 2837.53 | 2816.1 | 2816.1 |
| | 11 | 54 | 2819.3 | 2804.7 | 2883.4 | 2825.48 | 2804.7 | 2804.7 |
| | 20 | 90 | 2819.3 | 2789.0 | 2873.5 | 2819.75 | 2789.0 | 2789.0 |
| | 30 | 124 | 2819.3 | 2774.3 | 2867.2 | 2815.33 | 2774.3 | 2774.3 |
| | 41 | 148 | 2819.3 | 2763.8 | 2858.4 | 2808.62 | 2763.8 | 2763.8 |
| | 49 | 165 | 2819.3 | 2755.3 | 2849.7 | 2800.07 | 2755.3 | 2755.3 |
| | 60 | 194 | 2819.0 | 2740.8 | 2842.7 | 2794.45 | 2740.8 | 2740.8 |
| HC fire-20 mm FI | 5 | 50 | 2819.3 | 2806.5 | 2883.3 | 2839.45 | 2806.5 | 2806.5 |
| | 10 | 179 | 2819.3 | 2748.6 | 2856.5 | 2806.73 | 2748.6 | 2748.6 |
| | 20 | 459 | 2334.9 | 2376.5 | 2407.6 | 2336.41 | 2302.4 | 2248.2 |
| | 30 | 636 | 1197.5 | 1460.8 | 1155.9 | 701.58 | 1110.3 | 745.1 |
| | 40 | 754 | 729.2 | 767.6 | 626.5 | 291.80 | 591.2 | 299.4 |
| | 51 | 871 | 514.9 | 336.0 | 430.1 | 125.86 | 330.7 | 110.5 |
| | 60 | 954 | 428.6 | 187.1 | 333.6 | 124.16 | 238.7 | 93.0 |
| ISO834-30 mm FI | 5 | 24 | 2819.3 | 2817.3 | 2892.9 | 2851.94 | 2817.3 | 2817.3 |
| | 10 | 39 | 2819.3 | 2810.9 | 2886.9 | 2842.53 | 2810.9 | 2810.9 |
| | 20 | 70 | 2819.3 | 2797.4 | 2876.4 | 2833.12 | 2797.4 | 2797.4 |
| | 29 | 99 | 2819.3 | 2785.0 | 2869.4 | 2822.57 | 2785.0 | 2785.0 |
| | 41 | 130 | 2819.3 | 2771.5 | 2853.2 | 2812.02 | 2771.5 | 2771.5 |
| | 50 | 164 | 2818.6 | 2756.0 | 2840.8 | 2802.80 | 2756.0 | 2756.0 |
| | 60 | 270 | 2813.7 | 2703.1 | 2827.0 | 2784.13 | 2703.1 | 2703.1 |
| External fire-30 mm FI | 5 | 25 | 2819.3 | 2817.2 | 2888.2 | 2851.04 | 2817.2 | 2817.2 |
| | 10 | 39 | 2819.3 | 2811.2 | 2886.5 | 2844.35 | 2811.2 | 2811.2 |
| | 19 | 65 | 2819.3 | 2799.8 | 2877.7 | 2835.50 | 2799.8 | 2799.8 |
| | 30 | 92 | 2819.3 | 2788.1 | 2869.7 | 2826.64 | 2788.1 | 2788.1 |
| | 40 | 113 | 2819.3 | 2779.0 | 2861.6 | 2821.47 | 2779.0 | 2779.0 |
| | 54 | 136 | 2819.3 | 2768.9 | 2857.8 | 2816.16 | 2768.9 | 2768.9 |
| | 60 | 143 | 2819.3 | 2765.8 | 2854.9 | 2810.26 | 2765.8 | 2765.8 |
| HC fire-30 mm FI | 5 | 39 | 2819.3 | 2811.2 | 2887.0 | 2846.17 | 2811.2 | 2811.2 |
| | 10 | 134 | 2819.3 | 2769.8 | 2863.4 | 2818.41 | 2769.8 | 2769.8 |
| | 20 | 324 | 2819.3 | 2672.3 | 2789.7 | 2753.56 | 2672.3 | 2672.3 |
| | 30 | 493 | 2151.4 | 2217.2 | 2147.9 | 2004.16 | 2038.9 | 1797.8 |
| | 40 | 619 | 1304.9 | 1555.4 | 1291.5 | 938.84 | 1206.8 | 829.5 |
| | 51 | 717 | 855.0 | 987.7 | 857.5 | 504.52 | 741.6 | 416.2 |
| | 60 | 777 | 677.8 | 670.6 | 672.0 | 319.34 | 524.9 | 250.0 |

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
