# Peer review of "Inelastic Strength for Fire Resistance of Composite I-Beam Covered by Insulation Material Subjected to Basic Loading Condition"

_metals, doi:10.3390/met11050739_

Round 1

Reviewer 1 Report

The originality and the scientific value of the subject research can be better.

The research area is Inelastic Strength for Fire Resistance of Composite I-Beam Covered by Insulation Material Subjected to Basic Loading Condition.

There is extensive research in the area of ​​application of the finite element method and fire resistance. It is essential to rework the introduction chapter. 

Sucharda, O. et. al. Recommendation for the modelling of 3D non-linear analysis of RC beam tests. Comput. Concr. 2018, 21, 11–20.
Lee, S.-H.; Choi, B.-J. Post Fire Residual Strength of the Wall-Slab Using Siliceous Concrete. Materials 2021, 14, 1793.

In the part - Structural Analysis, very little information is given about the numerical model, input parameters for the calculation and setting of the FEM solver. More detailed information is required.

It is possible to use the chosen concept for numerical modeling, but the detail, acceptability, and information value are insufficient. 

It is necessary to document in detail the performed calculations and results, including the parameters of the calculation, the sensitivity of the calculation, and also the output. This part must be fundamentally reworked and expanded.
It would be appropriate to perform a sensitivity study and not only select (limited) calculations.
State the material parameters for concrete and steel in the detailed table.

The discussion needs to be reworked in a fundamental way. New knowledge must be clearly stated in the context of current knowledge.

The manuscript must be fundamentally reworked or resubmitted.

Author Response

Response to Reviewer’s Comments

Reviewer 2 Report

The manuscript is very interesting and presents an approach that attracts
the reader, the use of modern systems for practical situations, such as fire,
are very relevant for structures based on composite materials. I suggest some minor corrections: a) The abstract, although well written, lacks the presentation of the main results of this research,it must be inserted between one or two lines; b) In the introduction and review of the literature, I suggest to the authors an approach on composite materials, mainly reinforced materials (such as concrete)
and other types of reinforcement, which will help to solve another problem in this manuscript, which are the low number of references, only 30, of which several
are technical standards. I suggest the insertion of the following works: 10.1016 / j.jobe.2020.101675 10.1520 / JTE20200656 10.1016 / j.cscm.2020.e00406 10.1016 / j.conbuildmat.2021.123059 10.3969 / j.issn.0258-2724.20190244 10.1007 / s12649-021- 01374-5 10.1016 / j.conbuildmat.2021.123136 http://www.rlmm.org/ojs/index.php/rlmm/article/view/720%20-%20OPOGRAPHICAL%20SURFACE%20ANALYSIS%20OF%20PASTE%20AND%20CONCRETE%20MADE% 20WITH% 20BLENDED% 20COMMERCIAL% 20CEMENT all of these jobs must be considered. c) Reference 30 must be revised! d) "The properties of the mixed beam are 43A grade steel with a yield strength of 255 MPa and 30 grade concrete with 30 MPa compressive strength".
Could you give more details at this point? e) Item 6.4 must be rewritten and improved, more information can be added. In general, the authors are to be congratulated for the work, and I look forward to these small adjustments.

Author Response

Response to Reviewer’s Comments

Round 2

Reviewer 1 Report

The research area and results are from the context of the manuscript can better understand.

Thanks for the comments and manuscript edits.

The manuscript has sufficient informational value.

The presentation of the research and results is also at a good level.

The manuscript can be accepted for publication.